# Periostin and Thymic Stromal Lymphopoietin—Potential Crosstalk in Obstructive Airway Diseases

**DOI:** 10.3390/jcm9113667

**Published:** 2020-11-15

**Authors:** Patrycja Nejman-Gryz, Katarzyna Górska, Magdalena Paplińska-Goryca, Małgorzata Proboszcz, Rafał Krenke

**Affiliations:** Department of Internal Medicine, Pulmonary Diseases and Allergy, Medical University of Warsaw, Banacha 1a, 02-097 Warsaw, Poland; drkpgorska@gmail.com (K.G.); mpaplinska@wum.edu.pl (M.P.-G.); m.proboszcz@wp.pl (M.P.); rkrenke@wum.edu.pl (R.K.)

**Keywords:** asthma, induced sputum, periostin, TSLP

## Abstract

Periostin and thymic stromal lymphopoietin (TSLP) are newly described markers of obstructive airway diseases and the mechanism by which both markers participate in immune response remains poorly understood. The aim of our study was to determine periostin and TSLP concentration in serum and induced sputum (IS) in patients with atopic asthma, chronic obstructive pulmonary disease (COPD), and controls, as well as to evaluate the potential link between periostin, TSLP, and Th2 immune response. Serum and IS levels of periostin, TSLP, IL-4, and IL-13 were determined in 12 atopic asthmatics, 16 COPD sufferers, and 10 controls. We noticed a significantly higher IS periostin and TSLP concentration at protein and mRNA level in asthmatics compared to the two other groups; additionally, periostin and TSLP were correlated positively with IS eosinophil count. A strong positive correlation between IS periostin and TSLP protein levels (r = 0.96) as well as mRNA expression level (r = 0.95) was found in patients with asthma. The results of our study show that periostin and TSLP are associated with eosinophilic airway inflammation and seem to be important drivers of atopic asthma but not COPD pathobiology. Very strong correlations between local periostin, TSLP, eosinophils, and IL-4 in asthma point to the link between periostin–TSLP and Th2 response.

## 1. Introduction

Periostin (a member of protein fasciclin family) is involved in various biological processes such as cell proliferation, cell invasion, and tissue development, repair, and remodeling [1,2]. The role of periostin has been extensively studied in patients with bronchial asthma in which periostin regulates subepithelial fibrosis and mucus production [3]. However, it must be admitted that there are still significant gaps in our knowledge on periostin function in the lung. Periostin is secreted by bronchial epithelial cells and pulmonary fibroblasts and is recognized as a biomarker of type 2 eosinophilic inflammation [4,5]. Its production is upregulated by IL-4 and IL-13, which is characteristic for a Th2-type immune response [3]. Besides asthma, periostin is also involved in other diseases, such as atopic dermatitis, allergic rhinitis, and chronic rhinosinusitis [6]. Despite its seemingly relevant clinical significance, the mechanism by which periostin participates in immune response in chronic inflammatory airway diseases remains poorly understood.

Thymic stromal lymphopoietin (TSLP), an epithelial derived cytokine, primes and stimulates dendritic cell (DC) maturation and enhances the recruitment of Th2 effector cells by influencing DC–T cell crosstalk (including Th2 cytokine expression-induced eosinophil recruitment and IgE switching) [7]. The highest levels of TSLP are found in lung and skin-derived epithelial cells, but it can also be produced by fibroblasts, granulocytes, and DCs [8]. TSLP production and release from airway epithelial cells is associated with mechanical damage and the effects of inflammatory cytokines (TNF, IL-1α, IL-4, and IL-5) or proteases [9]. TSLP plays an important role in airway responses to allergens in patients with allergic asthma as well as in maintaining airway eosinophilic inflammation in these subjects [10,11]. 

A possible connection between periostin and the TSLP pathway in atopy and Th2-type inflammation has been previously reported. It was observed that periostin enhances the pathogenesis of atopic dermatitis (AD) through TSLP production from keratinocytes in an AD mouse model and in vitro co-culture system of keratinocytes [12]. In vitro analyses revealed that periostin promoted survival and proliferation of keratinocytes and directly induced production of TSLP. Takahashi et al. implied that periostin together with TSLP play an important role in creating a Th2-dominant environment in the development of cutaneous T-cell lymphoma (CTCL), and reported that the levels of periostin and TSLP correlated with the level of IL-4 [13]. The authors showed elevated TSLP and periostin mRNA levels in CTCL skin lesions. They also found that dermal fibroblasts isolated from these lesions abundantly expressed periostin at both mRNA and protein levels when stimulated with IL-4 or IL-13. Other authors demonstrated that increased periostin expression in the oral mucosa and serum of patients with oral lichen planus (OLP) was associated with inflammatory response, where Th2 cytokine was predominant in immune imbalance and elevated TSLP concentrations [14]. Upgraded levels of periostin and TSLP in serum, induced sputum (IS), and bronchoalveolar lavage fluid (BALf) have been reported in patients with asthma and chronic obstructive pulmonary disease (COPD) [15,16,17,18,19]. However, to our best knowledge, there have been no studies on the simultaneous assessment of periostin and TSLP levels and their potential link with the Th2 immune response in patients with these two obstructive lung diseases.

Taking into account the above data, we undertook a study to determine periostin and TSLP concentrations in the serum and induced sputum of patients with atopic asthma and COPD, as well as to evaluate the potential link between periostin, TSLP, and Th2 immune response in these two airway diseases. The specific aims of the study were to (a) measure serum periostin and TSLP concentrations; (b) assess periostin and TSLP levels in IS at the protein and mRNA levels; (c) compare the obtained values in atopic asthma, COPD, and controls; and (d) search for potential correlations between periostin, TSLP, and other biomarkers of Th2 immune response in the asthma, COPD, and control group.

## 2. Experimental Section

### 2.1. Overall Study Design

Thirty-eight subjects who were patients with either atopic asthma or COPD, or healthy subjects, were enrolled into this prospective cross-sectional study. Patients were recruited from the outpatient clinic of the Department of Internal Medicine, Pulmonary Diseases and Allergy, Medical University of Warsaw, Poland between 2012 and 2016. The comprehensive medical history with particular attention to data on the onset of asthma/COPD symptoms, disease duration and severity, history of atopy, and smoking were collected from all subjects. All study participants underwent lung function testing with bronchial reversibility (when applicable) and methacholine challenge test, allergy skin prick test (SPT), blood sampling for basic laboratory investigations (including blood eosinophils and total IgE level). Serum and IS from all patients were collected and the levels of periostin, TSLP, IL-4, IL-13 were determined at the protein level by ELISA method in these materials. In IS cells, the mRNA expression for the preselected cytokines was evaluated using the qPCR method. The study protocol was approved by the Bioethics Committee of the Medical University of Warsaw, Poland (KB/186/2010). All subjects provided written informed consent.

### 2.2. Study Subjects and Disease Definitions

The diagnosis and severity of asthma were established according to the Global Strategy for Asthma Management and Prevention (GINA) [20]. The major diagnostic criteria for asthma were as follows: medical history of episodic breathlessness, wheezing, cough, and chest tightness; spirometric features of airway obstruction with positive bronchial reversibility test and/or a positive result of methacholine challenge test. Patients with mild-to-moderate disease scored according to GINA 2012 were included.

The diagnosis and severity assessment of COPD were based on the Global Initiative for Chronic Obstructive Lung Disease (GOLD) report [21]. The following criteria had to be met as a condition for enrollment into the COPD group: signs and symptoms consistent with COPD, a positive smoking history (>10 pack-years), persistent airway obstruction (FEV1/FVC < 70%) in post-bronchodilator spirometry, GOLD stage 1–2 severity of airway obstruction. Two common inclusion criteria for both asthma and COPD patients comprised (1) stable disease defined as no exacerbations requiring treatment escalation and lack of respiratory infection for at least 6 weeks before the study enrolment, and (2) no systemic or inhaled steroid use 6 weeks prior to study onset.

The control group consisted of subjects with no medical history of obstructive lung diseases, with normal spirometry results.

### 2.3. Lung Function Parameters and Atopy Status

In all patients, lung function was assessed by spirometry (Lungtest 1000, MES, Cracow, Poland) with bronchial reversibility testing (when applicable). The measurements were performed in accordance with the recommendations of the European Respiratory Society (ERS) and the American Thoracic Society (ATS) [22,23]. A positive bronchodilator response was defined as an increase of >200 mL and ≥12%, of the predicted value, in either FEV1 or FVC. The methacholine challenge test was performed consistently according to ATS guidelines [24], between one to seven days before sputum induction.

Atopy was defined as a positive skin prick test (3 mm in diameter in the presence of positive histamine and negative diluent controls) to at least one of 15 extracts of common local aeroallergens [25]. Total IgE concentration was evaluated in serum using ELISA Biomerieux mini Vidas (Marcy-l’_Etoile, France) according to the manufacturer’s instruction (measurement range 0.5–1000 kIU/L).

### 2.4. Sputum Induction and Processing

Sputum induction was preceded by inhalation of four puffs of salbutamol (400 μg) and subsequent spirometry. Induction was performed with sterile hypertonic saline (NaCl) at increasing concentrations (3%, 4%, and 5% solutions) via an ultrasonic nebulizer (ULTRA-NEBTM2000, DeVilbiss, Port Washington, NY, USA) in accordance with the ERS guidelines [26]. Spirometry was repeated after each inhalation. The induction was stopped after a decrease in FEV1 by at least 20% from the baseline (post bronchodilator) value. Sputum plugs were isolated from saliva and were processed with 0.1 % solution of dithiothreitol (DTT, Sigma Aldrich, St. Louis, MO, USA). After centrifugation, the obtained supernatants were stored at −70 °C for cytokine measurements and the cell pellet was resuspended in RNAlater solution (Qiagen, Hilden, Germany) and stored at −80 °C for further cytokine mRNA investigation.

The total cell count (10^6^ cells/g sputum) was determined in Neubauer hemocytometer with particular attention to the number of epithelial cells (the sample was considered inappropriate when epithelial cells accounted for >50% of the detected cells). Two smears from each subject were stained using the May–Grunwald–Giemsa method to assess the percentage of different immune cells, based on the morphology of 300 cells from various fields.

### 2.5. RNA Isolation and cDNA Synthesis

Total RNA was isolated from IS pellet cells using NucleoSpin RNA II Columns Kit (Machery & Nagel, Düren, Germany) according to the manufacturer’s instruction. The RNA was treated with DNAse to remove genomic DNA before reverse transcribing into cDNA. The concentration and purity of isolated RNA was measured on a spectrophotometer (Beckman DU650, Krefeld, Germany) using the 260/280 nm absorbance ratio. Total RNA (5 μL) was used for reverse transcription with GoScript Reverse Transcriptase for qPCR (Promega Corporation, Madison WI, USA).

### 2.6. Real-Time Quantitative PCR

Real-time quantitative PCR evaluation (qPCR) was performed using an ABI-Prism 7500 Sequence Detector System (Applied Biosystems, Foster City, CA, USA). Expression of mRNA for selected cytokines was assayed using TaqMan reagents. Primers and probes for the cytokines periostin Hs01566734_m1, TSLP Hs00263639_m1, IL-4 Hs00174122_m1, and IL-13 Hs00174379_m1 were commercially available from ThermoFisher (ThermoFisher Scientific, Waltham, MA USA). qPCR was performed in 25 μL reaction volumes containing TaqMan Universal Master Mix (Thermo Fisher Scientific, Waltham, MA, USA) with 150 nM of specific primers and 100 nM of FAM/TAMRA-labeled probe with the standard TaqMan temperature profile (2 min at 50 °C, 10 min at 95 °C, and 40 cycles of 15 s at 95 °C and 1 min at 60 °C). mRNA copy numbers were calculated for each sample by using the cycle threshold (Ct) value. All samples were amplified in triplicate, and the mean was obtained for further calculations. Before starting the appropriate experiments, the reference gene 18S rRNA was experimentally selected from several reference genes. 18S rRNA was used as an internal control for each sample to normalize the gene expression level. The mRNA expression of the control group was used as a calibrator. The relative gene expression was calculated using the comparative 2^−ΔΔCT^ method where ΔΔCt = ΔCT target group − ΔCT control group. The results were expressed as relative quantification (RQ) units (fold change).

### 2.7. Cytokine Concentration Measurements

Cytokine levels (periostin, TSLP, IL-4, IL-13) were measured in serum and IS supernatants from all study participants using kits for standard quantitative sandwich enzyme-linked immunosorbent assay (ELISA) technique according to the manufacturer’s instructions. All samples were determined in duplicate. We used Periostin/OSF-2 human ELISA kit (Phoenix Pharmaceuticals, Burlingame, CA, USA) with a sensitivity of 0.027 ng/mL and detection range of 0.027 to 20 ng/mL; Human TSLP ELISA kit (Invitrogen, Carlsbad, CA, USA) with a sensitivity of 3 pg/mL and detection range of 3.3–800 pg/mL; Human IL-4 High Sensitivity ELISA (IBL International, Switzerland) with a sensitivity of 0.25 pg/mL and detection range of 0.25–16 pg/mL; and IL-13 Human ELISA kit (Thermo Fisher Scientific, Waltham, MA, USA) with a sensitivity of 0.7 pg/mL and detection range of 1.6–100 pg/mL.

In the preliminary study, the evaluation of any possible effects of DTT on the cytokines during sputum processing was performed. The evaluated level of protein standards reconstituted in PBS–DTT dilution buffer did not differ from those dissolved in the recommended diluents; therefore, the final protein measurements were made without DTT addition to the standard.

### 2.8. Statistical Analysis

Estimation of the sample size was calculated on the basis of our earlier study in which periostin concentration in respiratory samples was analyzed [15]. To detect whether the difference in periostin level between the investigated groups was similar to those previously reported with a power of 80% and significance level of 5%, the sample size was estimated as a minimum of 10 subjects in each group.

Statistical analysis was performed with the use of Statistics 12.0 software package (Stat Soft Inc., Tulsa, OK, USA). Data were expressed as medians and interquartile ranges (IQRs) (25–75th percentiles) or numbers and percentages. Quantitative data distribution was assessed using Shapiro–Wilk test. Comparisons between the groups were made using nonparametric Mann–Whitney test or ANOVA Kruskal–Wallis test. Categorical variables were compared using the chi-squared test or Fisher’s exact test. To evaluate the significance of correlation coefficient, Spearman correlation test was used. Statistical significance was accepted at a *p*-value lower than 0.05.

## 3. Results

### 3.1. Patients Characteristics

There were 12 patients with mild-to-moderate atopic asthma, 16 with mild-to-moderate COPD (GOLD 1–2 severity of airway obstruction), and 10 controls. Basic demographic data and clinical and functional characteristics of the study participants are shown in Table 1.

Patients with COPD were older than asthmatics and controls, but the difference did not reach statistical significance. Asthma patients had the highest level of total serum IgE and sputum and serum eosinophils, while COPD patients were characterized by the highest sputum neutrophils count (Table 1). Asthma and COPD patients had significantly impaired lung function compared to the control group. Significantly greater impairment of respiratory function (lower FEV1% predicted and FEV_1_/FVC%) was demonstrated in COPD than in the other studied groups. In the asthma group, 10 out of 12 patients had positive results of reversibility test, while in the COPD group, all patients had negative results for this test. Airway hyperresponsiveness was found in 10 out of 16 COPD patients who underwent methacholine challenge.

### 3.2. Periostin, TSLP, IL-4, and IL-13 mRNA Expression and Protein Levels in Serum and Sputum in Control, Asthma, and COPD Group

Concentrations of the investigated cytokines in serum and IS supernatants, as well as their mRNA expression in sputum cells, are presented in Table 2.

In general, the levels of all the investigated cytokines were higher in serum than in IS. The smallest (2-fold) difference between serum and IS concentrations was found for TSLP. The fold differences for the remaining cytokines ranged between 6 and 185.

There were no significant differences in serum cytokine concentrations between the three investigated groups; however, the absolute concentrations of periostin and TSLP were higher in patients with asthma compared to patients with COPD and controls (without attaining statistical significance) (Table 2). By contrast, a significantly higher IS periostin and TSLP level in asthmatics compared to the two other groups was demonstrated. The highest serum IL-4 and IL-13 levels were found in asthma patients, but the intergroup differences were not significant (Table 2). Significant differences in sputum periostin and TSLP mRNA expressions were shown, with the highest values in asthma and the lowest in controls (Table 2).

Subanalysis performed within the asthma group (IgE > 100/ IgE < 1 00 kIU/L; sputum eosinophilia > 3%/sputum eosinophilia < 3%) did not show any significant differences in terms of tested cytokine concentrations at protein and mRNA level. 

### 3.3. Correlations between Cytokine mRNA and Protein Levels and Sputum Cellular Profile

In all three investigated groups, a strong significant correlation was found for sputum periostin concentration at protein and mRNA expression levels: r = 0.98, *p* < 0.001 for asthma; r = 0.91, *p* < 0.001 for COPD; and r = 0.79, *p* < 0.001 for controls. Such a correlation was also observed for TSLP in patients with either asthma or COPD (r = 0.95 and r = 0.64, respectively, *p* < 0.001 for both), but not in controls. There were no correlations between protein and mRNA levels of IL-4 and IL-13. In patients with asthma, both IS concentration and mRNA expression of periostin and TSLP correlated positively with IS eosinophil count (Table 3). This was also the case for IS periostin but not IS TSLP concentration and eosinophil count in patients with COPD. There was no correlation between sputum neutrophils and IS periostin or IS TSLP in any of the investigated groups. However, significant positive correlations between both IS IL-4 and IL-13 protein level and IS eosinophil count in the asthma group were observed (r = 0.76, *p* = 0.006 and r = 0.68, *p* = 0.02, respectively). We did not find such correlations for the COPD and control group.

A strong positive correlation between IS periostin and TSLP protein levels (r = 0.95, *p* < 0.001) as well as mRNA expression level (r = 0.95, *p* < 0.001) was found in patients with asthma (Figure 1). There were neither significant correlations between serum periostin and TSLP levels in asthma nor IS or serum periostin and TSLP levels (or their mRNA expression) in the COPD and control groups (Figure 1).

Additionally, we found a strong positive correlation between periostin and IL-4 and IL-13 concentrations, as well as for TSLP with IL-4 and IL-13 at the IS protein level in asthma patients (Figure 2), with no correlation at the mRNA level.

### 3.4. Relationships between Sputum Eosinophilia and Increased TSLP and Periostin Concentrations in Control, Asthma, and COPD Groups

Subanalysis carried out in patients with and without atopy, and patients with vs. without high serum IgE level showed no significant differences in concentrations of periostin and TSLP at both protein and mRNA level.

The analysis of coexistence of sputum eosinophilia (>3%), high sputum periostin, and high sputum TSLP concentration (above the median value of the control group), showed that almost 64% patients with asthma had increased sputum eosinophilia with concomitant elevated levels of IS periostin and TSLP (Figure 3). By contrast, only 15% of COPD patients and 0% in the control group had elevated sputum eosinophilia with high periostin and TSLP levels at the same time.

Most patients with concomitantly elevated levels of sputum eosinophils, periostin, and TSLP also had also increased IL-4 and IL-13 concentrations (6/7 patients for both cytokines, which represents 50% of the asthma group). In the COPD group, only one person had simultaneously raised levels of sputum eosinophils, periostin, TSLP, IL-4, and IL-13.

## 4. Discussion

Our study showed that both periostin and TSLP were associated with eosinophilic airway inflammation and seemed to be important drivers of atopic asthma pathobiology. By contrast, the role of these mediators in COPD was incomparable—weaker or even none. To our best knowledge, this is the first study aimed at concurrent measurement of serum and IS periostin and TSLP levels in patients with different obstructive lung diseases. To date, levels of only periostin or only TSLP have been studied in asthma and COPD patients [2,3,8,19]. We found that two-thirds of mild-to-moderate atopic asthma patients who were not treated with ICC had IS eosinophilia together with relatively high IS periostin and TSLP levels. Moreover, we believe that the results of our study can be considered as an initial but solid ground for pointing out a new phenomenon—positive crosstalk between these two cytokines in asthma, but not COPD.

In general, the results of the study are consistent with existing data on elevated serum and sputum periostin concentration in asthma [27,28,29,30,31] and COPD patients [31] compared to healthy controls. Similarly to other studies, we found that periostin level in asthmatics is higher than in COPD patients [32,33]. In terms of TSLP, the results of our study are also in agreement with previously reported higher serum and sputum levels in asthmatics [17,34,35] vs. healthy subjects, as well as COPD patients [18,36]. Alike to the other authors, we confirmed that not only periostin and TSLP levels but also their IS mRNA levels were significantly higher in asthmatics than in control subjects [34,37,38,39]. It has previously been reported that smoking is an additional factor affecting TSLP levels in COPD patients [40]. This phenomenon may explain the elevated levels of TSLP in our COPD patients, 44% of whom were current smokers.

We believe that the relationship between IS periostin, TSLP, and IS cellular composition is an important point in our study. A significant positive correlation of IS periostin at both protein and mRNA levels with sputum eosinophilia in asthmatics is in line with the earlier observations [29,41]. These findings seem to confirm that periostin can be a biomarker of airway eosinophilia in asthmatic patients and has a potential utility in patient selection for asthma therapeutics targeting Th2 inflammation. Although serum periostin has been recognized to be related with eosinophilic inflammation in asthma [29,42], its role in COPD has not yet been elucidated. Studies investigating the correlation between periostin and eosinophils in COPD patients are scarce. There are some data indicating a positive correlation between serum periostin and serum eosinophils [43], but we could not find any reports on sputum the eosinophil–periostin relationship in COPD. Our study showed a significant positive correlation between IS periostin concentration and IS eosinophils in the COPD group. This finding may highlight that some COPD patients develop inflammation with a Th2 immunologic component and may require a different treatment approach. This seems to be in line with the “treatable traits” concept of obstructive lung diseases management [44]. A significant positive correlation between IS TSLP level and IS eosinophils in asthma patients, demonstrated in our study, might somehow be related to results of the study by Wong et al. These authors reported that human eosinophils constitutively expressed the functional heterodimeric receptor for TSLP [11]. We can speculate that TSLP acts indirectly on eosinophils through innate lymphoid cells (ILC2) that can promote IL-5-mediated eosinophil recruitment, which was suggested by other authors [45]. Another important mechanism for TSLP-mediated eosinophilia in skin and the bronchial airway is probably due to the local production of the eosinophil chemokine eotaxin-2 by TSLP-activated DCs [11]. It should be underlined that in our three study groups, only patients with asthma were characterized by a very strong positive correlation between IS periostin and TSLP concentrations both at the protein and mRNA levels (r = 0.96 and r = 0.95, respectively). Similar data describing the interaction of periostin and TSLP have been described in AD [12], CTCL [13], and OLP [14]. This might be related to their likeness during inflammatory reaction, because each of these diseases is associated with chronic inflammation, where the Th2 immune response is dominant.

The results of our study suggest that there might at least be local interplay between periostin and TSLP in airway inflammation in atopic asthma. TSLP produced by the epithelium might represent a crucial factor involved in allergic stimulation and allergic response at epithelial surfaces [46]. It has been demonstrated that TSLP activates dendritic and mast cells, affecting the production of cytokines associated with Th2-type response [47] and recruitment of Th2 cells to the lung [48]. Periostin is another factor that is produced by airway epithelial cells but also by fibroblasts and mast cells, mainly as a result of IL-4 and IL-13 stimulation [3]. Kim et al. found a positive correlation between periostin levels and the number of integrin αV-positive epithelial cells in patients with eosinophilic nasal polyps [49]. In their study, both periostin and integrin αV expression levels were correlated with TSLP production. Thus, the authors concluded that the periostin–integrin αV pathway plays a role in TSLP production in airway epithelial cells. The secretion of TSLP by airway epithelial cells and DCs may be induced by ligands (bacteria, viruses) that activate toll-like receptors (TLR) through nuclear factor NF-κB activation [50]. Moreover, it has been reported that periostin can activate NF-κB signaling, thereby increasing IL-6 production [51]. Hence, NF-κB might be a common regulator of the periostin–integrin αV pathway. One early study showed that NF-κB and STAT6 can interact directly and synergistically activate IL-4-induced genes and activators of NF-κB [52]. Kim et al. suggested that periostin might enhance synergistic activity of Th2 cytokine-induced STAT6 and NF-κB to induce TSLP in eosinophilic nasal polyps [49].

We may speculate that similar link between periostin and TSLP action occurs in atopic asthma, where periostin produced by MCs can affect epithelial cells via integrin binding activation, resulting in TSLP secretion. In contrast to asthma, we did not find a relationship between periostin and TSLP in the COPD and control groups. Additionally, we noted that 7/12 (58%) patients with asthma had a relatively high level of IS periostin together with sputum eosinophilia and a relatively high level of IS TSLP. This leads us to suppose the existence of a link between periostin–TSLP and the Th2 response as asthma is mainly caused by Th2 inflammation with the predominance of mast cells and eosinophils in airways [7]. High sputum TSLP levels may be induced by overexpressed periostin in the airway epithelium, but the specific mechanisms underlying the concurrent increase in local TSLP and periostin levels in asthma need further investigation in in vitro and in vivo experimental studies.

Our study has several limitations. The major flaw is the relatively small numbers of patients with asthma and COPD who participated in the study. Obviously, confirmation of the results and more in-depth analysis of the relationship between periostin, TSLP, and cellular characteristics of the airway inflammation in larger studies are warranted. Recruitment may have been somewhat limited by the important exclusion criterion used in our study, i.e., treatment with medications which could affect the course of airway inflammation—this includes ICS. The second limitation is related to the fact that the patients with only two severity stages (mild and moderate) of asthma and COPD were evaluated. This was not only due to a higher risk of complications in patients with more severe stages of the disease but also to the significant difficulty in finding and enrolling patients with more severe asthma who would not be treated with ICS. The next limitation is associated with the differences in the cigarette smoke exposure in our study groups. As the asthma and control groups included mainly non-smokers, it is difficult to compare their results with active smokers in COPD group. It is known that smoking can affect TSLP production, and the results in COPD groups should be rather compared to those in healthy smokers. Another point is that TSLP results are quite close to the limit of detection of used kit. We decided to show this data because for all subjects in the control group, the concentration of TSLP in the sputum did not exceed 3 pg/mL, while in the other two groups, there were only single cases (1 in the asthma and 3 in the COPD group).

## 5. Conclusions

The results of our study show that periostin and TSLP are associated with eosinophilic airway inflammation and seem to be important drivers of asthma but not COPD pathobiology. Moreover, a very strong correlation between local periostin and TSLP level suggests positive crosstalk between these two cytokines in asthma. The relationships between periostin, TSLP, eosinophils, and IL-4 in asthma point to the links between periostin–TSLP and the Th2 response. Thus, the periostin–TSLP axis may be a potential future therapeutic target in asthma.

## Figures and Tables

**Figure 1 jcm-09-03667-f001:**
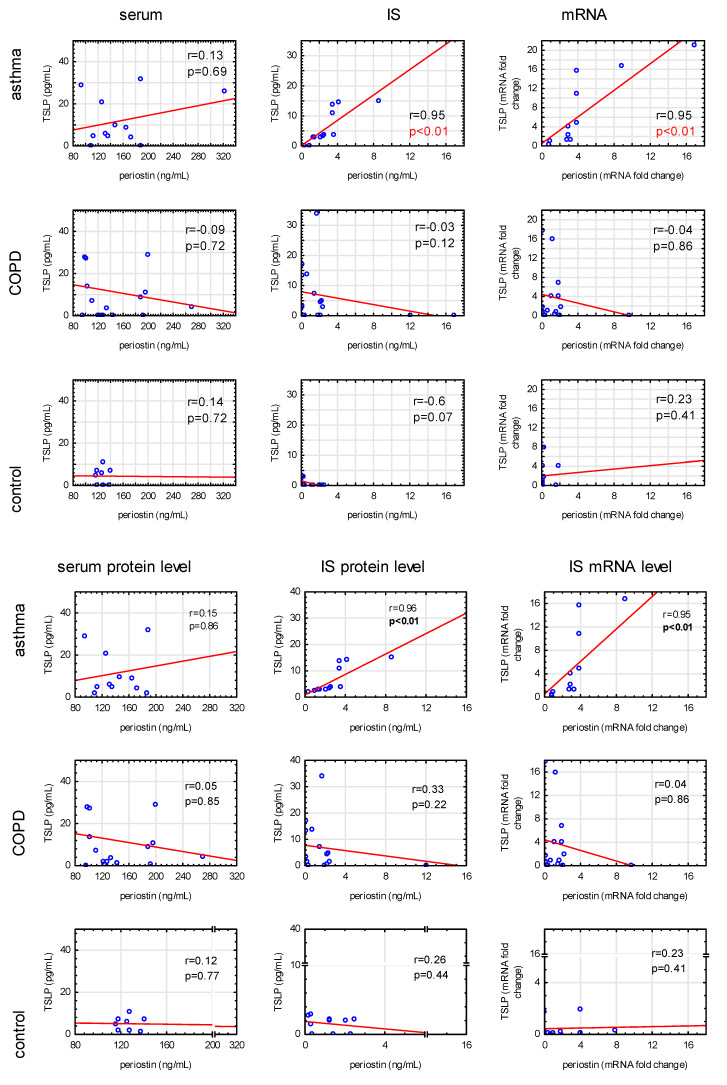
Correlations between periostin and TSLP concentration in serum, induced sputum, and at mRNA level in asthma, COPD, and control subjects; r—Spearman’s rank correlation coefficient.

**Figure 2 jcm-09-03667-f002:**
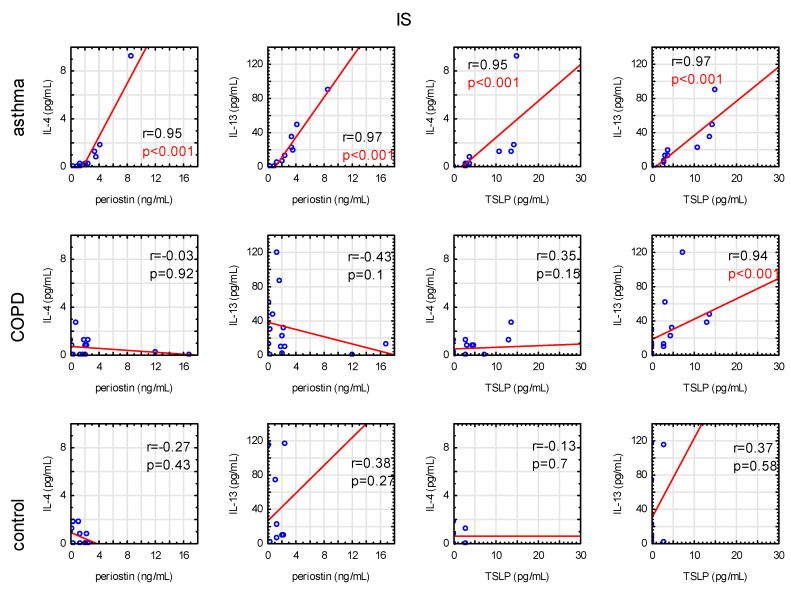
Correlations between concentrations of periostin or TSLP with IL-4 or IL-13 in induced sputum in asthma, COPD, and control subjects; r—Spearman’s rank correlation coefficient.

**Figure 3 jcm-09-03667-f003:**
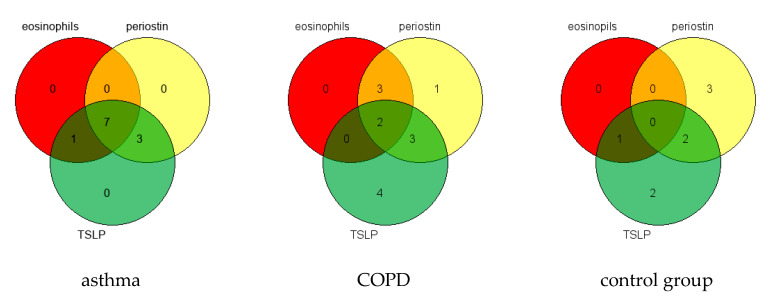
Venn diagrams showing the distribution of asthma, COPD, and healthy patients with increased sputum eosinophils count and elevated levels of IS periostin and IS TSLP.

**Table 1 jcm-09-03667-t001:** Baseline characteristic of asthma, chronic obstructive pulmonary disease (COPD), and control subjects.

	Asthma (n = 12)	COPD (n = 16)	Control (n = 10)	*p*-Value
**Demographic Characteristics**				
Age (years)	43.5 (27–59)	65 (60–74.5)	40.5 (29–63)	0.14
Sex (M/F)	7/5	10/6	3/7	0.25
BMI (kg/m^2^)	27.4 (25.5–31)	27.9 (24–30.1)	26.2 (20.7–29.8)	0.67
Current/ex-/never smokers, n (%)	0/3/9 (0/25/75)	7/9/0 (44/56/0)	1/2/7 (10/20/70)	**0.0002**
Smoking history (pack/years)	0 (0–0.25)	45 (34–50)	2 (0–14)	**0.001**
Disease duration (months)	56.5 (33–108)	48 (24–120)	n/a	0.73
Atopy status (atopic/non-atopic)	12/0	6/10	5/5	0.07
**Pulmonary Function (Prebronchodilator Values)**				
FEV_1_ (% predicted)	92.5 (82–106)	71.5 (67.5–82)	106.5 (99–109)	**0.001**
FVC (% predicted)	113 (98.5–127)	103 (95.5–110)	111.5 (108–118)	**0.03**
FEV_1_/FVC (%)	67 (62.5–72.5)	56 (51.5–60.5)	76 (70–83.6)	**0.001**
Airway hyperresponsiveness (PC_20_ mg/mL)	0.2 (0.08–0.5)	0.6 (0.3–10.1)	20 (14.2–20)	**0.001**
**Selected Laboratory Parameters**				
Serum total IgE (kIU/L)	144 (76–238.5)	40.5 (7–176.5)	36.5 (26–77.5)	**0.05**
Positive result of skin prick-tests, n (%)	6 (4–10.5)	2 (0–4)	2 (0–5.5)	0.071
Blood eosinophils (%)	4 (2–6.5)	2 (1–3.5)	2.5 (1.5–4)	0.11
Blood eosinophils (× 10^9^/L)	0.2 (0.17–0.38)	0.15 (0.07–0.19)	0.16 (0.09–0.22)	0.15
Blood neutrophils (%)	56 (44–61.5)	59.5 (52.5–64)	59.5 (51.5–63.5)	0.6
Blood neutrophils (× 10^9^/L)	3.3 (2.5–4.7)	4.4 (3–5.4)	3.4 (2.8–4.1)	0.55
Sputum eosinophils (%)	10 (1–57)	1 (0.5–3.5)	1 (0–2)	**0.001**
Sputum neutrophils (%)	25 (21–38)	75.5 (56.5–92.5)	52.5 (43–64)	**0.006**

Data are presented as median and interquartile ranges (IQR); *p*-values were obtained using Kruskal–Wallis test or Fisher’s exact test. BMI—body mass index, FEV1—forced expiratory volume in one second, FVC—forced vital capacity.

**Table 2 jcm-09-03667-t002:** Serum and sputum periostin, TSLP, IL-4, and IL-13 expression in asthma, COPD, and control subjects.

	Asthma (n = 12)	COPD (n = 16)	Controls (n = 10)	*p*-Value
**Serum Cytokine Level**				
Periostin (ng/mL)	141.3 (119.2–180)	126.2 (107.1–190.5)	126.7 (118.4–132.9)	0.6
TSLP (pg/mL)	7.3 (4.5–23.3)	5.4 (0–20.3)	4.9 (0–6.9)	0.69
IL-4 (pg/mL)	46.3 (12.1–83.4)	15.3 (8.3–21.7)	10.4 (1.9–17.8)	0.23
IL-13 (pg/mL)	224.8 (92.7–385)	143.8 (107.1–204.7)	60.3(21.4–102.5)	0.08
**Sputum Cytokine Level**				
Periostin (ng/mL)	2.5 (1.4–3.5)	1.6 (0.26–2.2)	1.23 (0.32–2.04)	**0.049**
TSLP (pg/mL)	3.5 (3–12.3)	3.08 (0–10.2)	0 (0–0)	**0.01**
IL-4 (pg/mL)	0.25 (0.125–1.25)	0.25 (0–1.25)	0.3 (0–1.25)	0.76
IL-13 (pg/mL)	12 (4.5–28.25)	22 (9.5–47)	9.5 (2–74.5)	0.71
**Sputum mRNA Cytokine Expression**				
Periostin (fold change)	3.1 (1.8–3.9)	1.3 (0.3–1.9)	1 (0.4–3.9)	**0.04**
TSLP (fold change)	3.1 (1–13.2)	0.95 (0.08–4)	0.16 (0.1–0.24)	**0.01**
IL-4 (fold change)	0.6 (0.15–3.9)	0.05 (0.01–14.8)	0.17 (0.05–111.7)	0.53
IL-13 (fold change)	0.005 (0–0.09)	0.002 (0.001–0.003)	10.1 (4.6–12.8)	0.06

Data are presented as median and interquartile ranges (IQR); *p*-values were obtained using Kruskal–Wallis test.

**Table 3 jcm-09-03667-t003:** Correlation of periostin and TSLP concentration with eosinophils in asthma, COPD, and control subjects.

	Serum Periostin (ng/mL)	IS Periostin (ng/mL)	mRNA Periostin Expression	Serum TSLP (pg/mL)	IS TSLP (pg/mL)	mRNA TSLP Expression
**Asthma**						
Blood eosinophils (%)	r = 0.08 *p* = 0.8	r = 0.1 *p* = 0.73	r = 0.05 *p* = 0.87	r = 0.28 *p* = 0.36	r = −0.02 *p* = 0.96	r = 0.11 *p* = 0.72
IS eosinophils (%)	r = 0.27 *p* = 0.41	r = 0.67 *p* = **0.024**	r = 0.67 *p* = **0.02**	r = 0.14 *p* = 0.66	r = 0.73 *p* = **0.01**	r = 0.75 *p* = **0.007**
**COPD**						
Blood eosinophils (%)	r = −0.07 *p* = 0.79	r = 0.36 *p* = 0.38	r = 0.36 *p* = 0.38	r = 0.2 *p* = 0.44	r = 0.16 *p* = 0.53	r = −0.08 *p* = 0.77
IS eosinophils (%)	r = 0.12 *p* = 0.65	r = 0.49 *p* = **0.048**	r = 0.5 *p* = 0.057	r = −0.17 *p* = 0.52	r = −0.15 *p* = 0.67	r = 0.02 *p* = 0.93
**Controls**						
Blood eosinophils (%)	r = 0.36 *p* = 0.38	r = 0.28 *p* = 0.49	r = 0 *p* = 1	r = 0.2 *p* = 0.61	r = 0.18 *p* = 0.63	r = −0.3 *p* = 0.45
IS eosinophils (%)	r = 0.15 *p* = 0.72	r = −0.23 *p* = 0.51	r = −0.54 *p* = 0.1	r = −0.02 *p* = 0.95	r = 0.17 *p* = 0.63	r = −0.32 *p* = 0.36

r—Spearman’s correlation coefficient.

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
