# Peer review of "Periostin and Thymic Stromal Lymphopoietin—Potential Crosstalk in Obstructive Airway Diseases"

_jcm, 2020, doi:10.3390/jcm9113667_

Round 1

Reviewer 1 Report

This paper by Patrycja Nejman-Gryz and colleagues describes that periostin and TSLP are associated with eosinophilic airway inflammation and seem to be important drivers in asthma, but not COPD pathobiology. This provides the first description of the linkage between periostin, TSLP, and eosinophilic inflamattion. These results are novel and very exciting and will provide important new insight into , which will be of broad interest to many in the field of pathophysiology and treatment of asthma. Overall, the manuscript is well written and the conclusions largely supported. Some thoughts and comments to improve the paper are provided below:

Major points
1, Fowler said that a blood eosinophil count of 450 / ul or higher predicts a sputum eosinophil count of 2% or higher (Stephen J Fowler, et al. High blood eosinophil counts predict sputum eosinophilia in patients with severe asthma. J Allergy Clin Immunol. 2015), so please indicate the actual number of blood eosinophils.

2, Methacholine airway hyperresponsiveness is said to be positive at 8 mg or less (Sumino K, et al. Methacholine challenge test: Diagnostic characteristics in asthmatic patients receiving controller medications. J Allergy
Clin Immnol 2012.), some of patients with COPD also seemed to appear to have airway hyperresponsiveness, is there any consideration for that?

Minor points

There is a description about airway reversibility in the Experiment Section, so please display in the results or not.

Author Response

Response to Reviewer 1 Comment

Dear Reviewer

Thank you very much for your review of the manuscript and for taking the time to read it. Thanks for your comments and suggestions that will help to improve understanding of the manuscript.

  1. Fowler said that a blood eosinophil count of 450 / ul or higher predicts a sputum eosinophil count of 2% or higher (Stephen J Fowler, et al. High blood eosinophil counts predict sputum eosinophilia in patients with severe asthma. J Allergy Clin Immunol. 2015), so please indicate the actual number of blood eosinophils

We agree with this comment and have incorporated your suggestion throughout the manuscript. Data on the number of blood eosinophils have been added to the Table 1. in section: Selected laboratory parameters. Additionally, in the same place Table1 we also wrote the total number of neutrophils in the blood.

Line number 201 after change: “Asthma patients had the highest level of total serum IgE and sputum and serum eosinophils, while COPD patients were characterized by the highest sputum neutrophils count (Table 1).”

In our group of patients with asthma, the number of eosinophils in the blood were lower than in the cited SJ Fowler publication, which may be due to the fact that asthmatics groups differ in the degree of severity of the disease. Fowler's population was described by the group of patients with severe asthma, while our patients have the mild-moderate form of the disease.

We wanted to focus on the local response rather than the peripheral inflammation which is a feature of severe asthma and COPD rather than the mild-to-moderate stages, so we decided to analyze the level of cytokines and number of eosinophils in induced sputum, not serum.

  1. Methacholine airway hyperresponsiveness is said to be positive at 8 mg or less (Sumino K, et al. Methacholine challenge test: Diagnostic characteristics in asthmatic patients receiving controller medications. J Allergy Clin Immnol 2012.), some of patients with COPD also seemed to appear to have airway hyperresponsiveness, is there any consideration for that?

Thank you for pointing this out. In accordance with the literature data airway obstruction and airway hyperresponsiveness (AHR) are important features of asthma and frequently observed in COPD.  Both diseases are characterized by airway wall and lung tissue inflammation, and in asthma there exists a relationship between the inflammatory state of the airways and the severity of hyperresponsiveness. However, the type and cause of this inflammation, as well as the extent and consequences of the inflammatory process, are different in asthma and COPD. Inflammatory processes affecting the airway wall both in peripheral and central areas of the lung appear to be important, the former one dominating in COPD and the latter in asthma. Many individuals with COPD demonstrate increased responsiveness as well.

In our study airway hyperresponsiveness was found in 10 out of 16 COPD patients who underwent the methacholine challenge. We obtained similar data in our previous studies where nearly 75% of COPD patients had airway hyperresponsiveness (Proboszcz M et al. Relationship between Blood and Induced Sputum Eosinophils, Bronchial Hyperresponsiveness and Reversibility of Airway Obstruction in Mild-to-Moderate Chronic Obstructive Pulmonary Disease. COPD: Journal of Chronic Obstructive Pulmonary Disease. 2019; 16:5-6, 354-361).

The occurrence of AHR in COPD is influenced by multiple mechanisms, among which impairment of factors that oppose airway narrowing plays an important role. The main determinants of AHR in COPD are reduction in lung function and smoking status.

The sentence: The airway hyperresponsiveness was found in 10 out of 16 COPD patients who underwent the methacholine challenge. was added to Patients characteristics section (line 205-209)

  1. There is a description about airway reversibility in the Experiment Section, so please display in the results or not.

We have added this information in the manuscript in Result section in line number 207: “In the asthma group 10 out of 12 patients had positive results of reversibility test, while in COPD group all patients had negative results of this test”.

Reviewer 2 Report

This study compared levels of type 2 cytokines( IL-4, IL-13, TSLP, and periostin)  in sera and induced sputa of patients with adult asthma, COPD, and healthy controls. However, several issues were raised as listed below.

  1. The study number of each group: The authors enrolled only 12 asthmatics, 16 COPD, and 10 controls which are not enough to compare cytokine and biomarker levels in two kinds of samples., therefore statistical analysis within 12 asthmatic patients are not reliable.
  2. Asthmatic patients: higher periostin levels were noted in patients with severe asthma with type 2 phenotypes. However, you enrolled patients with a mild-to-moderate degree and did not compare whether type 2 cytokines and periostin levels were different according to the phenotypes of type 2 responses, therefore, pl analyze these points. In addition, you mentioned that to define asthmatic patients, the GINA2012 was applied. Why you apply this old guideline? The most recent one is the GINA2020 guidelines.
  3. Cytokine levels in induced sputum: It is not easy( is impossible) to collect sputum samples from healthy controls. DTT could degrade protein/cytokines within the sputum samples, therefore you have to confirm these values you presented are real values or not. In addition, type 2 cytokine levels in sera are very low.  In order to measure periostin levels, the results may be different according to the ELISA Kits used. Pl try to measure these levels using different kits. 
  4. Pl provide the rationale on how to select candidate cytokines of epithelial cells among TSLP, IL-33 and others. 

Author Response

Dear Reviewer,

I would like to thank you very much for reviewing our manuscript and pointing out the points that we should change to make the results clearer. We tried to address all remarks and comments. According to the recommendations I introduced changes to the manuscript (by using the "Track Changes" function in Microsoft Word). To make the text easier for readers, we have introduced the names of the subsections in the Results Section. In addition, we have re-edited figure 1 to make it clearer.

  1. The study number of each group: The authors enrolled only 12 asthmatics, 16 COPD, and 10 controls which are not enough to compare cytokine and biomarker levels in two kinds of samples., therefore statistical analysis within 12 asthmatic patients are not reliable.

We realize that the small size of the study groups is a weak point of our research. This was due to several reasons: the requirement of not using steroids for 6 weeks prior to the study onset greatly limited the number of enrolled patients, particularly asthmatics; not all qualified patients managed to perform procedure of sputum induction;  in the case of several patients we found that sputum contains too many epithelia and was considered as not meeting the previously established criteria (epithelial count <50%).

Despite the small size of the groups, it seems to us that the obtained results are interesting and worth presenting in the form of preliminary studies. Obviously, confirmation of our results and a more in-depth analysis of the relationship between periostin, TSLP and the cellular characteristics of airway inflammation require further studies on a larger number of patients.

  1. Asthmatic patients: higher periostin levels were noted in patients with severe asthma with type 2 phenotypes. However, you enrolled patients with a mild-to-moderate degree and did not compare whether type 2 cytokines and periostin levels were different according to the phenotypes of type 2 responses, therefore, pl analyze these points. In addition, you mentioned that to define asthmatic patients, the GINA2012 was applied. Why you apply this old guideline? The most recent one is the GINA2020 guidelines.

Thank you for this suggestion. We have added “atopic” word to asthma to emphasise the point that all our patients were atopic, patients with dominant Th2 phenotype of asthma. (line 14,17,24,73,79,193,287,292,334,352).

We performed additional statistical calculations to check whether the concentrations and the mRNA level of the tested cytokines differed in the asthma subgroups separated on the basis of the amount of IgE, the amount of positive skin tests, and the number of eosinophils in the sputum. The performed analyzes did not show any differences in concentration of evaluated cytokines on protein and mRNA level.

The sentence: “Subanalysis performed within the asthma group (IgE>100/IgE<100; sputum eosinophilia >3%/ sputum eosinophilia <3%) did not show any significant differences in terms of tested cytokines concentration at protein and mRNA level” was added in the Results section line 230.

We used the GINA 2012 guidelines because we started collecting material from the first patients in 2012. Patients were recruited into the project and classified as mild-moderate asthma based on these guidelines.

  1. Cytokine levels in induced sputum: It is not easy( is impossible) to collect sputum samples from healthy controls. DTT could degrade protein/cytokines within the sputum samples, therefore you have to confirm these values you presented are real values or not. In addition, type 2 cytokine levels in sera are very low.  In order to measure periostin levels, the results may be different according to the ELISA Kits used. Pl try to measure these levels using different kits. 

We know that performing the sputum induction in healthy people is difficult but possible. Our center has experience in performing sputum induction. Our center has many years of experience in performing sputum induction, as evidenced by numerous studies based on induced sputum in various patient groups, including healthy volunteers (Górska K et al; Comparison of cellular and biochemical markers of airway inflammation in patients with mild-to-moderate asthma and chronic obstructive pulmonary disease: an induced sputum and bronchoalveolar lavage fluid study. J.Physiol. Pharmacol. Off. J. Pol. Physiol. Soc. 2008; 59 Suppl 6: 271–283. IF 2,631; Górska K et al Comparative study of periostin expression in different respiratory samples in patients with asthma and chronic obstructive pulmonary disease. Pol. Arch. Med. Wewn. 2016; 126: 124–137; PapliĹ„ska-Goryca M et al Expression of Inflammatory Mediators in Induced Sputum: Comparative Study in Asthma and COPD. Advances in Experimental Medicine and Biology 2016; 1–1; Górska K et al Comparative Study of IL-33 and IL-6 Levels in Different Respiratory Samples in Mild-to-Moderate Asthma and COPD. COPD- Journal of Chronic Obstructive Pulmonary Disease. 2018; 15: 36-4; PapliĹ„ska-Goryca M et al Sputum interleukin-25 correlates with asthma severity: a preliminary study. PostÄ™py Dermatologii i Alergologii. 2018; 35:462-469).

We agree that DTT can degrade protein by destruction of disulfide bridges, however, the use of DTT is essential to dissolve the mucus in induced sputum. Therefore, we conducted preliminary experiments in which we checked the impact of DTT on cytokine concentration in the sputum. The addition of DTT did not affect the results to any great extent We have included the obtained results in the Experimental section line 174-177.

In the preliminary study, the evaluation of any possible effects of DTT on the cytokines during sputum processing was performed. The level of evaluated protein standards reconstituted in PBS-DTT dilution buffer did not differ from those solutes in recommended diluents; the final protein measurements were made without DTT addition to the standard.”

As for the cytokine concentrations in the tested materials, indeed the sputum level is much lower than the serum level. This may be due, first, to the fact that sputum reflects local processes in the respiratory system and this material is diluted by inhaled NaCl solution. The serum is a mixture of cytokines from various components of the whole body, we see the inflammatory response on a general level. Secondly, cytokine ELISA kits are dedicated to serum, there are no kits available specifically for sputum, which may also be important. The most important thing is that we compare the samples marked with one common ELISA kit. So, if sputum concentrations are lower than serum concentrations, they are equally lower in all groups tested.

We are aware that cytokine concentrations measured with different ELISA kits may differ, but we follow the principle to measure the cytokine level in samples using one company’s kit during one experiment so that the results are comparable. Unfortunately, we are not able to measure the concentration of periostin with other available ELISA kit, due to the use of the whole materials collected from patients.

A few years ago we published a paper comparing TSLP determination with different ELISA kits: Górska K, Nejman-Gryz P, PapliĹ„ska-Goryca M, Proboszcz M, Krenke R. Comparison of Thymic Stromal Lymphopoietin Concentration in Various Human Biospecimens from Asthma and COPD Patients Measured with Two Different ELISA Kits. Advances in Experimental Medicine and Biology. 2017; 955:19–27). The results obtained with the EIAab kit were 3 to even 45-fold higher than those measured with the R&D kit. Significant differences between the investigated groups were found only for the TSLP concentration in induced sputum. We conclude that TSLP level is highly dependent on the ELISA kit used for the measurement. Thus, judgement on TSLP results obtained with different assays might be confusing and lead to wrong conclusions. Probably a similar situation would be the case with other cytokines.

  1. Pl provide the rationale on how to select candidate cytokines of epithelial cells among TSLP, IL-33 and others. 

Based on our previous experience, where we tested levels of IL-6 and Il-33 in different materials from patients with asthma and COPD, we knew that IL-33 did not differentiate between these patient groups. IL-33 levels reached similar values in asthma and COPD in all investigated samples- serum, induced sputum, and exhaled breath condensate. (Górska K, Nejman-Gryz P, PapliĹ„ska-Goryca M,KorczyĹ„ski P, Prochorec-Sobieszek M, Krenke R. Comparative Study of IL-33 and IL-6 Levels in Different Respiratory Samples in Mild-to-Moderate Asthma and COPD. COPD- Journal of Chronic Obstructive Pulmonary Disease. 2018; 15: 36-45).

We drew similar conclusions about the IL-25- we have published results for sputum IL-25 at various asthma severities. Our results suggest although IL-25 in IS did not differ between patients with asthma and healthy subjects is particularly associated with severe asthma. (Paplińska-Goryca M, Grabczak EM, Dąbrowska M, Hermanowicz-Salamon J, Proboszcz M, Nejman-Gryz P, Maskey-Warzechowska M, Krenke, R. Sputum interleukin-25 correlates with asthma severity: a preliminary study. Postępy Dermatologii i Alergologii. 2018; 35:462-469). In our experiments described in this manuscript, we included patients with mild-moderate asthma, so we did not expect any differences in IL-25 levels between the study groups.